# Optimum k-Nearest Neighbors for Heading Synchronization on a Swarm of UAVs under a Time-Evolving Communication Network

**DOI:** 10.3390/e25060853

**Published:** 2023-05-26

**Authors:** Rigoberto Martínez-Clark, Javier Pliego-Jimenez, Juan Francisco Flores-Resendiz, David Avilés-Velázquez

**Affiliations:** 1Faculty of Engineering, Administrative, and Social Sciences, Autonomous University of Baja California, Tecate 21460, BC, Mexico; 2Electronics and Telecommunications Department, Applied Physics Division, CICESE-CONACYT, Ensenada 22860, BC, Mexico

**Keywords:** k-nearest neighbors, time varying network, heading synchronization, flocking

## Abstract

Heading synchronization is fundamental in flocking behaviors. If a swarm of unmanned aerial vehicles (UAVs) can exhibit this behavior, the group can establish a common navigation route. Inspired by flocks in nature, the k-nearest neighbors algorithm modifies the behavior of a group member based on the k closest teammates. This algorithm produces a time-evolving communication network, due to the continuous displacement of the drones. Nevertheless, this is a computationally expensive algorithm, especially for large groups. This paper contains a statistical analysis to determine an optimal neighborhood size for a swarm of up to 100 UAVs, that seeks heading synchronization using a simple P-like control algorithm, in order to reduce the calculations on every UAV, this is especially important if it is intended to be implemented in drones with limited capabilities, as in swarm robotics. Based on the literature of bird flocks, that establishes that the neighborhood of every bird is fixed around seven teammates, two approaches are treated in this work: (i) the analysis of the optimum percentage of neighbors from a 100-UAV swarm, that is necessary to achieve heading synchronization, and (ii) the analysis to determine if the problem is solved in swarms of different sizes, up to 100 UAVs, while maintaining seven nearest neighbors among the members of the group. Simulation results and a statistical analysis, support the idea that the simple control algorithm behaves like a flock of starlings.

## 1. Introduction

Unmanned aerial vehicles (UAVs), commonly called drones, are versatile mobile robots that have captivated the attention of the automatic control community. UAVs have different topologies, there exist fixed-wing ones, which resemble airplanes and, in recent years, multi-rotor systems have become the standard for UAV tasks. For example, UAVs have been employed for pesticide and fertilizer dissemination, seed planting, and weed recognition, etc. [1]. Other applications include search and rescue missions, cinematography, and photogrammetry, among others [2].

Regarding photogrammetry, UAVs can aid in the texturization of 3D buildings in the creation of urban models, especially for urban planning or architectural design. In [3], an algorithm for the selection of the optimal building facade texture image, obtained from oblique images gathered with a camera array mounted on a drone, is presented. On the other hand, in [4] the authors used a drone assisted camera array, along with an algorithm based on Gaussian process regression and quantization of the polynomial mutation operator, to solve the many-objective optimization problem. This algorithm targets four optimization criteria: coverage rate, point-level clarity, uniform clarity, and utilization rate. The authors relied on the high mobility and flexibility offered by the UAVs, to create the camera array.

Using different sensors to acquire information remotely, is called remote sensing. This activity has benefited from the characteristics of UAVs, and in the past decade, marine research and monitoring has exploited it with myriad applications, ranging from marine disaster and environmental monitoring, such as algal and jellyfish blooms or man-made disasters like oil spills, to marine surveying and mapping management, such as coastline measurement; see [5] and references therein for further applications.

A further problem to which groups of UAVs can be applied, is communication networks. Using these robots, in combination with 5G technology, the coverage area for communication networks can be increased [6], and on-demand communications services can be offered in temporary massive events or in disaster relief [7]. UAVs will play a key role in smart cities. Traffic management and road safety could also rely on this technology [8]. Another popular future application employing UAVs, will be the delivery of goods [9].

Most of the aforementioned tasks, would benefit from a multi-agent scheme. The advantages of this scheme include robustness, expanded area coverage, simultaneous task completion, and power management. Various control techniques for multi-agent systems have been applied to UAV groups. In [10], the authors employed an event-triggered distributed model predictive control, in order to achieve coordination among a group of UAVs. On the other hand, a master–slave scheme is widely employed, with fixed roles [11] or dynamic leader changing [12]. Moreover, in [13] the authors designed a control strategy based on a fractional-order recurrent neural network, combined with an interval-valued intuitive fuzzy system, in order for a group of unmanned underwater vehicles to decide its maneuvering strategy in a dynamic target scenario. This combination allowed the group to take into account the enemy dynamic in a counter-game strategy, while dealing with the uncertainties due to the environment, such as weak connectivity or high perturbances. Nevertheless, the bulk of multi-agent systems control is based on graph theory. This generally involves fixed communication graphs.

A current trend in multi-agent systems, is the use of swarm robotics. This is a bio-inspired control scheme, that takes its inspiration from the self-organized behavior of social organisms, thus, does not require a declared leader, exhibits high flexibility, including time-evolving communication networks, and has increased robustness [14]. Some control techniques for robotic swarms, include ant colony optimization, which is based on the pheromone trails left by ants to reinforce a singular option of a mutual exclusive set [15,16,17]. On the other hand, particle swarm optimization, is an algorithm that represents interactions in self-organized groups with evolution capabilities. In this case, the members of the group share their best solution to the problem, and other members compare it with theirs, and a cost function determines which they take forward to the next generation of solutions [18,19,20].

Another technique, is the so called k-nearest neighbors (k-NN) algorithm, mainly employed in classification problems, in which, a candidate solution is compared with a training set, to determine the nearest possible outcome [21]. Regarding swarm robotics, this technique was inspired by the flocking behavior of birds. In this collective behavior, birds modify their travel orientation based on the orientation of their flockmates [22]. The main drawback of this scheme, is that the algorithm has $O(n^{2})$ complexity, because in every timestep, each member of the group has to determine its neighborhood [23]. The *k*-NN algorithm, is a strategy to determine a subset of the group, to apply another consensus or synchronization strategy. This is important, because this allows the information flow on the network to be reduced and eases the calculations on every drone. For this reason, it is important to determine a suitable synchronization technique. Among the simplest control strategies, the so called consensus algorithm for continuous system is employed widely in the literature. This is a P-like algorithm frequently used in first-order dynamic systems, and relies on graph theory to function [24].

In order for this strategy to be able to achieve a consensus, it is important that the communication graph contains a spanning tree [25]. Consensus has successfully been achieved when implementing this strategy in [26], for a group of micro air vehicles exhibiting a formation behavior. In [27], the authors extended the same consensus protocol to a second-order system, obtaining a control strategy for a 3D formation in a group of UAVs. The authors claim that the quick response of the algorithm and high control precision, make this a suitable controller for engineering applications. On the other hand, in [28] the authors tackled the formation problem with a group of fixed-wing UAVs, employing fuzzy model reference adaptive control, to promote self-tuning of a vector-field-based consensus protocol. This combination allows target tracking with the group, while exhibiting an adaptive circular motion around the objective. Furthermore, the authors of [29] combined an improved artificial potential field strategy, with the second-order consensus protocol, to ensure formation and obstacle avoidance in a group of double integrator model UAVs. This enhancement in the potential field, can move attractive fields to the desired position, create repulsive fields for static or dynamic obstacles, and take into account the effect of other UAVs creating another potential field with them.

Another important collective behavior that is related to consensus or synchronization, is flocking. This behavior is present in nature, and many collective organisms can exhibit it, such as schools of fish, flocks of birds, or herds of land mammals. The flocking behavior is employed to avoid predators, increase the efficiency of locomotion, or even as a hunting tactic. This collective behavior is characterized by a fluid movement of the whole group, resembling a single entity [30]. A few decades ago, Reynolds extracted the pillar behaviors of flocking, which are now considered as Reynolds’ flocking rules [31]:Cohesion, which means that the members of the group should stay together. This is achieved if the elements try to reduce the distance between them.Separation, the distance between elements should be enough to ensure that collisions between flockmates do not occur.Alignment, the members of the group have to match their speed vectors. This, has two requirements, they have to maintain similar linear velocity to stay in the flock, and the elements need to synchronize their orientation, in order to follow the same direction of the group.

In [32], the authors combined a vector field to obtain parallel formation, with an adaptive backstepping technique, to achieve flocking in a group of fixed-wing UAVs. The proposed strategy, allowed a leaderless convoy formation of the group. On the other hand, the authors of [33] employed a swarm-intelligence-based multi-layer flocking algorithm, to create a UAV communication network. Swarm-based strategies can optimize different criteria, with self-organization and self-adaptation. This strategy allows the group to fly in formation, avoiding collisions, but maintaining the quality of service of their provided signal, while optimizing power consumption. As stated before, another swarm inspired algorithm is particle swarm optimization (PSO). In [34], the authors proposed a flocking algorithm based on multi-objective PSO or MOPSO, this strategy allows the swarm to follow the Reynolds rules, nevertheless, the authors decided to train a neural network with the output of the MOPSO strategy, in order to approximate the mathematical relationship between the UAV flocking state and the flocking controller parameters, to reduce the calculation time of the former strategy. Continuing with bio-inspired techniques, in [35] the authors emulated the hierarchical leadership structure of homing pigeon flocks, to propose a flocking algorithm. In this scenario, multiple leader–follower relationships emerge, but at least two spanning trees emerge on the communication graph, this ensures that there are not isolated nodes in the group. Simulations with a swarm of fixed-wing UAVs, demonstrated the feasibility of the proposal.

As Reynolds’ rules establish, heading synchronization is fundamental to achieve flocking in a group of agents. One of the first models to cope with this problem, was proposed by Vicsek and coauthors [36], in which the particles reach a self-ordered motion, having constant velocity and changing their orientation with the average direction of motion of the particles in a set neighborhood. Nevertheless, this strategy has two main drawbacks: it has a singularity problem, due to the average, and it cannot guarantee global convergence of the heading of the group. To deal with these problems, in [37] the authors propose two different algorithms to achieve heading consensus. The first one, is a weighted version of Vicsek’s algorithm, which, along with a perturbation scheme, can cope with the singularity of the first algorithm. The second algorithm is based on a leader–follower scheme, in which there exists a member of the group that does not change its direction due to the influence of others. If the weighted algorithm is applied under this assumption, a heading reference is always present and, considering that a spanning tree exists in the communication graph, global convergence is achieved.

On the other hand, in [38] the authors decided to employ differential game theory to achieve heading consensus in a group of UAVs. In this case, their algorithm is based on a zero-sum game against a virtual target. This allows every UAV to minimize its heading error with the average of a neighborhood, according to a directed graph, while minimizing control efforts. The authors of [39], employed the basic consensus algorithm in a switching scheme, to achieve formation heading consensus, while the agents avoid obstacles and can perform in time-variant formations. this problem was solved by including a virtual graph, which performs the consensus and formation, and physical robots try to converge to one of the different virtual robots. Another benefit of this approach, is that the system can perform with robots of different kinematics, in the experimental results reported, they had a group of multiple-wheeled mobile robots combined with a UAV.

Considering a flocking scenario for a swarm of UAVs, with a simple heading synchronization algorithm and fixed travel velocity, this paper presents a statistical analysis to determine an optimum neighborhood size. As the UAVs travel around the workspace, they continuously change their neighbors, thus, the communication network employed to synchronize the orientation of the robots, is evolving through time. The remainder of the paper is organized as follows: Section 2 contains the mathematical model of the UAV and graph theory preliminaries, Section 3 presents the heading synchronization algorithm, while Section 4 establishes the statistical analysis, in order to find the optimum size of the neighborhood, along with the simulation results. Section 5 discusses the results, and finally, in Section 6 some conclusions are stated.

## 2. Preliminaries

Consider a group of UAVs, the simplified kinematics of a single drone are represented as [40]
(1)$$f\left(X,U\right)=\left(\begin{array}{c} \dot{\phi }\\ \dot{\theta }\dot{\psi }a_{1}+\dot{\theta }a_{2}\Omega_{r}+b_{1}U2\\ \dot{\theta }\\ \dot{\phi }\dot{\psi }a_{3}-\dot{\phi }a_{4}\Omega_{r}+b_{2}U_{3}\\ \dot{\psi }\\ \dot{\theta }\dot{\phi }a_{5}+b3U_{4}\\ \dot{z}\\ g-\left(\cos\phi \sin\theta \right)\frac{1}{m}U_{1}\\ \dot{x}\\ u_{x}\frac{1}{m}U_{1}\\ \dot{y}\\ u_{y}\frac{1}{m}U_{1} \end{array}\right)$$with
(2)$$\begin{array}{cc} a_{1}=\left(I_{yy}-I_{zz}\right)/I_{xx} & b_{1}=\ell /I_{xx}\\ a_{2}=J_{r}/I_{xx} & b_{2}=\ell /I_{yy}\\ a_{3}=\left(I_{zz}-I_{xx}\right)/I_{yy} & b_{3}=1/I_{zz}\\ a_{4}=J_{r}/I_{yy} & \\ a_{5}=\left(I_{xx}-I_{yy}\right)I_{zz} \end{array}$$
(3)$$\begin{array}{c} u_{x}=\left(\cos\phi \sin\theta \cos\psi +\sin\phi \sin\psi \right)\\ u_{y}=\left(\cos\phi \sin\theta \sin\psi -\sin\phi \cos\psi \right) \end{array}$$
with control inputs $U_{i}\;\;|\;\;i=1\ldots 4$.

If the drones are commanded to hover over a plane with fixed height, as seen in Figure 1, after some transformations, generally including backstepping control techniques [41], the kinematics of a single UAV can be expressed as the agent with nonholonomic constraints [42]:(4)$$\left[\begin{array}{c} \dot{x}\\ \dot{\theta } \end{array}\right]=\left[\begin{array}{cc} \cos\left(\theta \right) & 0\\ \sin\left(\theta \right) & 0\\ 0 & 1 \end{array}\right]\left[\begin{array}{c} \nu \\ \omega \end{array}\right]$$
where $x\in {ℜ}^{2}$ represents the position on the hovering plane, θ is the heading angle, and the inputs ν and ω correspond to the linear and angular velocities, respectively. If every UAV flies with a fixed linear velocity, the objective is to synchronize the heading angle θ, in order to drive a group to a common direction.

### Graph Theory

Consider a graph of the form G={N,E}, with the set of nodes N representing the group of UAVs and the set of edges E corresponding to the communication links between the members of N, is employed to model the group of UAVs. Some matrices are employed to define G.

The adjacency matrix Γ, contains the information of the communication streams, thus, this is a 0–1 matrix, whose elements are obtained as
(5)$$\gamma_{ij}=\{\begin{array}{cc} 1 & \mathrm{if}\;\{i,j\}\in E,\\ 0 & \mathrm{otherwise} \end{array}\;\;\;\mathrm{with}\;i\neq j\;\;\;$$

This implies that there are no weighted communications, i.e., if drones *i* and *j* are sharing information, $\gamma_{ij}=1$.

Another important matrix to consider, is degree matrix D, which is diagonal and sums all the elements that share information to UAV *i*, with regards to this, the elements of D are obtained as
(6)$$d_{ii}=\sum_{j=1,j\neq i}^{N}\gamma_{ij}$$

Lastly, the coupling matrix A, relates both matrices, to give a complete representation of a graph, because, for every node it contains the number of neighbors and which one is in the neighborhood. This matrix is obtained as follows
(7)A=Γ−D

For non-weighted communications, and considering there are not isolated elements, i.e., those which do not receive any information from other members of the group; and furthermore, if each UAV has exactly k-neighbors, all of the eigenvectors of A will be non-positive.

The following examples demonstrate how these matrices are obtained.

**Example 1.** 
*Consider the undirected graph of Figure 2, in this case, the information flows in both directions of every edge in the set E. The adjacency matrix *Γ* corresponding to this graph, is obtained as follows*

(8)
$$\Gamma =\left[\begin{array}{ccccc} 0 & 1 & 0 & 0 & 1\\ 1 & 0 & 1 & 0 & 1\\ 0 & 1 & 0 & 1 & 1\\ 0 & 0 & 1 & 0 & 1\\ 1 & 1 & 1 & 1 & 0 \end{array}\right]$$


*The degree matrix D is easily obtained from the sum of the elements in every row of the adjacency matrix:*

(9)
$$D=\left[\begin{array}{ccccc} 2 & 0 & 0 & 0 & 0\\ 0 & 3 & 0 & 0 & 0\\ 0 & 0 & 3 & 0 & 0\\ 0 & 0 & 0 & 2 & 0\\ 0 & 0 & 0 & 0 & 4 \end{array}\right]$$


*Recalling that the coupling matrix A is formed by the difference of the adjacency and degree matrices, therefore:*

(10)
$$A=\left[\begin{array}{ccccc} -2 & 1 & 0 & 0 & 1\\ 1 & -3 & 1 & 0 & 1\\ 0 & 1 & -3 & 1 & 1\\ 0 & 0 & 1 & -2 & 1\\ 1 & 1 & 1 & 1 & -4 \end{array}\right]$$



**Example 2.** 
*When information flows only in one direction through every edge, the graph is called a directed graph, or digraph. In Figure 3, the arrows in the edges indicate the direction of the information. For digraphs, the adjacency matrix takes into account the nodes from which node i can retrieve information. Equation (Equation 11) describes the adjacency matrix corresponding to the digraph of Figure 3*

(11)
$$\Gamma =\left[\begin{array}{ccccc} 0 & 1 & 0 & 0 & 0\\ 0 & 0 & 1 & 0 & 0\\ 0 & 0 & 0 & 0 & 1\\ 0 & 0 & 1 & 0 & 0\\ 1 & 1 & 0 & 1 & 0 \end{array}\right]$$


*The main difference between undirected graphs and digraphs, is that the adjacency, and consequently coupling, matrices are not symmetrical in the former case, a situation that occurs in matrices associated with undirected graphs. The degree matrix of the digraph in the example, takes the form*

(12)
$$D=\left[\begin{array}{ccccc} 1 & 0 & 0 & 0 & 0\\ 0 & 1 & 0 & 0 & 0\\ 0 & 0 & 1 & 0 & 0\\ 0 & 0 & 0 & 1 & 0\\ 0 & 0 & 0 & 0 & 3 \end{array}\right]$$


*In consequence, the coupling matrix of the digraph is formed as follows,*

(13)
$$A=\left[\begin{array}{ccccc} -1 & 1 & 0 & 0 & 0\\ 0 & -1 & 1 & 0 & 0\\ 0 & 0 & -1 & 0 & 1\\ 0 & 0 & 1 & -1 & 0\\ 1 & 1 & 0 & 1 & -3 \end{array}\right]$$



## 3. Heading Synchronization

Based on the simple control algorithm to synchronize groups of dynamical systems proposed by Wang [43], a control law for the *i*-th UAV, represented by (Equation 4), with the purpose of synchronization of the heading in a group of *N* UAVs, can take the form
(14)$$\omega_{i}=c\sum_{j\in N}a_{ij}\theta_{j}$$
where c>0 is a coupling gain. This corresponds to a proportional control law structure, in such a case, the synchronization state in a group of *N* UAVs is achieved asymptotically, as
(15)$$\theta_{1}\left(t\right)=\theta_{2}\left(t\right)=\cdots =\theta_{N}\left(t\right),\;\;\;\mathrm{as}\;t\to \infty$$

Lemma 1 of [44], establishes that the coupling gain, *c*, can be characterized by the largest non-zero eigenvalue of A. This implies, that synchronization can be achieved easily for high-degree graphs.

To assess this controller, a numerical simulation was performed. To achieve this goal, a script was developed in Matlab version R2018a, using a PC with Windows 10, 11th Gen Intel i5 processor, and 16 GB of RAM. The simulation scenario included a group of ten UAVs with nonholonomic constraints, described by (Equation 4), coupled with their five nearest neighbors. The initial positions and orientations were selected randomly over the workspace, and are depicted in Figure 4.

The experiment lasted 10 simulation seconds, with a sample period of 10 ms. The UAVs were set to travel at a fixed linear velocity of 15 m/s, over an obstacle-free plane, and their orientation was governed by the control law (Equation 14), with c=1. In every time step, all drones identified their 5-NN, in order to obtain $\omega_{i}\left(t\right)$, as stated before, due to the constant travel of the UAVs, the neighborhood of every UAV is constantly changing. After 1.5 s of simulation time, the orientation of the 10 UAVs was synchronized to 173.7 degrees, as shown in Figure 5. The RMS error of orientation between the drones, is used as a metric to evidence synchronization, and those errors tend to 0, as shown in Figure 6. The final positions of the UAVs are presented in Figure 7.

## 4. Optimum k-NN Neighborhood Size for a Group of UAVs

Observing the results from the previous example, it is clear that considering 50% of the group as a neighborhood is excessive, because the swarm synchronized their orientation easily. Thus, this percentage can be reduced to ease the complexity of the algorithm, especially if this scenario is intended to be applied to a swarm of UAVs with limited computational power. Inspired by this problem, two evaluation scenarios are proposed.

First, a statistical evaluation of the optimal percentage of neighbors in the swarm is conducted. The heading synchronization algorithm was repeated in a group of 100 UAVs, changing the neighborhood size in every experiment from 2% to 20%.

Second, and encouraged by [45], who established that neighborhoods in bird flocks are related to the density and not the spatial size of the flock. Furthermore, the entropy model of the authors determined that the neighborhood of the different European starling (*Sturnus Vulgaris*) flocks analyzed, was 6±0.6 birds. Thus, in the second experiment, the same heading synchronization algorithm was employed, but in this case, the group size was varied from 10 to 100 members, maintaining a neighborhood of seven UAVs.

Both experiments were realized 300 times, to acquire statistical relevance, and for both cases, the RMS heading error of the whole group was calculated and employed as the metric to determine if synchronization occurs. The simulation parameters were selected as the ones of the 10 UAV synchronization simulation described in the previous section. In all cases, the initial positions of the drones were selected randomly.

To obtain the RMS error, first, every drone calculated its mean absolute heading error $\theta_{e_{i}}$ with the other drones in the swarm, as
(16)$$\theta_{e_{i}}\left(t\right)=\frac{1}{N}\sum_{j=1,j\neq i}^{N}\left|\theta_{i}\left(t\right)-\theta_{j}\left(t\right)\right|$$

Then, the RMS error was obtained as the Euclidean norm of $\theta_{e}={\left(\theta_{e_{1}},\theta_{e_{2}},\cdots ,\theta_{e_{N}}\right)}^{T}$. Only the RMS error at t=10 s was used in the statistical analysis.

The boxplot of Figure 8, shows the RMS errors obtained at t=10 s for the 300 experiments applying the first scenario. It is clear that when over 10% of the group size is part of the neighborhood, synchronization is achieved. However, the median RMS error of an 8-NN, is 0.69 degrees, while the corresponding value for a 10-NN, is 0.115 degrees. In practice, we can consider a successful synchronization from 8-NN. Even more, the results for this particular neighborhood size are more concise, because they have fewer outliers than the 10-NN case.

In the second simulation environment with a fixed neighborhood size, from Figure 9 it can be seen that the median RMS heading error of the 300 experiments is below 1 degree for a swarm up to 80 UAVs, while for the bigger swarm, the median RMS heading error is 1.32 degrees.

## 5. Discussion

As observed in the heading synchronization simulation with ten UAVs and five neighbors, the goal was achieved within a few iterations of the algorithm. In this case, a 50% neighborhood size is excessive. With this in mind, the first analysis regarding the search for the optimal percentage of the group was performed. This experiment ranged from 2% to 20% neighborhood size of a 100-UAV swarm. The RMS error in the heading of the drones at the end of the experiment, was taken as the criterion to determine if the group achieved synchronization. For the 300 iterations run for every neighborhood size, from 8% and above, the results were satisfactory, this leads us to conclude that, less than 10% of the group in the neighborhood is sufficient to cope with the task.

Inspired by starling flocks, which have been reported to have around seven flockmates, a second experiment was performed. In this case, a neighborhood of seven UAVs was selected to apply the heading synchronization algorithm, with a swarm from 10 to 100 drones. In every case, the median RMS heading error of the 300 iterations per swarm size, was less than 1 degree, which in practice can be considered a successful synchronization, taking into account that a simple asymptotic control law was employed.

The scope of this work was the determination of an optimal neighborhood size for the k-NN algorithm, in that sense, only ideal conditions were considered. These include a full capacity of drones to exchange information with their swarm mates, without delay or distance limitation. Obviously, in real-world scenarios, these criteria should be taken into account. The *k*-NN algorithm does not define a sensing radius for the robots, it considers that a robot can communicate with its *k* neighbors. Thus, it is important to determine a good communication system, sensors with high capacity, if the intention is to apply the system in open space. The main obstacle of this algorithm is its computational cost, especially with large swarms, this highlights the importance of having an optimum neighborhood size, combined with simple control strategies. This leaves enough computational space to solve the flying algorithm, which can be adapted from the different options presents in the literature.

## 6. Conclusions

A statistical analysis for *k*-nearest neighbors heading synchronization in a swarm of UAVs, was developed in this paper. This problem is important, due to the high computational cost of the *k*-NN algorithm, in that sense, in order to reproduce at least 300 iterations of every experiment to obtain statistical significance, it was not possible to analyze swarms with more than 100 UAVs with the computational power used. Nevertheless, according to the literature, a flock of 500 starlings can synchronize using *k*-NN with k=7. Driven by this result, simulations employing one of the simplest synchronization algorithms found in the literature were performed. In one experiment, a 100-UAV swarm was evaluated, in which, after 300 iterations, a neighborhood of 8 to 10 neighbors was enough to achieve heading synchronization. On the other hand, in the experiment with a fixed 7-UAV neighborhood, evaluated in swarms from 10 to 100 drones, satisfactory results where obtained with up to an 80-UAV swarm. Probably, a better synchronization algorithm would ensure a full synchronization for a larger swarm. Nevertheless, *k*-NN is computationally expensive, thus, a simple control algorithm has to be employed. Future work can relate the size of the neighborhood with the coupling gain, *c*, to ensure that synchronization will be achieved, and in order to implement the solution, further analysis should be performed regarding communication capacity. Briefly, the conclusions of this work can be summarized as:There exists a bulk of literature on consensus, that relies on the simple P-like consensus algorithm, some of them combining it with another control technique to improve it benefits. This algorithm is efficient only if the communication graph is connected, or at least contains a spanning tree.A simple asymptotic P-like controller, was successfully employed to achieve heading synchronization on a group of up to 100 UAVs.For the different scenarios, the statistical evidence shows that seven neighbors were enough to cope with the problem, as observed in nature with flocks of European starlings.A small neighborhood, combined with a simple heading synchronization based on graph theory, is recommended to be employed with simple UAVs, especially for the swarm robotics community.Further analysis on the communication scheme should be made to implement this solution.

## Figures and Tables

**Figure 1 entropy-25-00853-f001:**
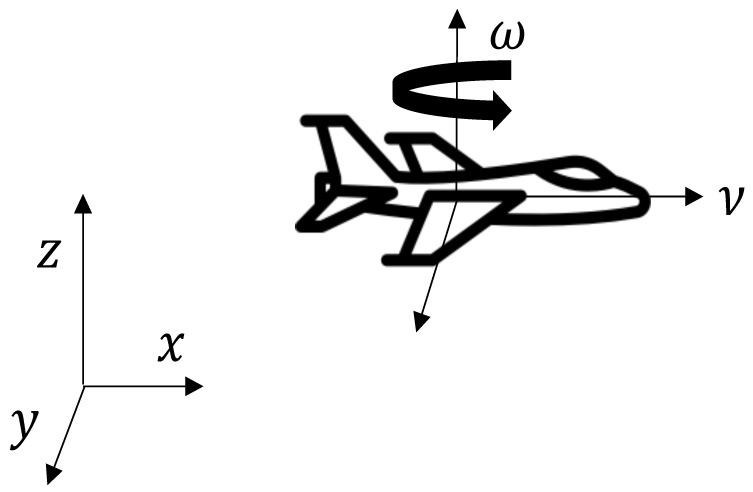
Kinematic representation of a UAV hovering, with fixed height.

**Figure 2 entropy-25-00853-f002:**
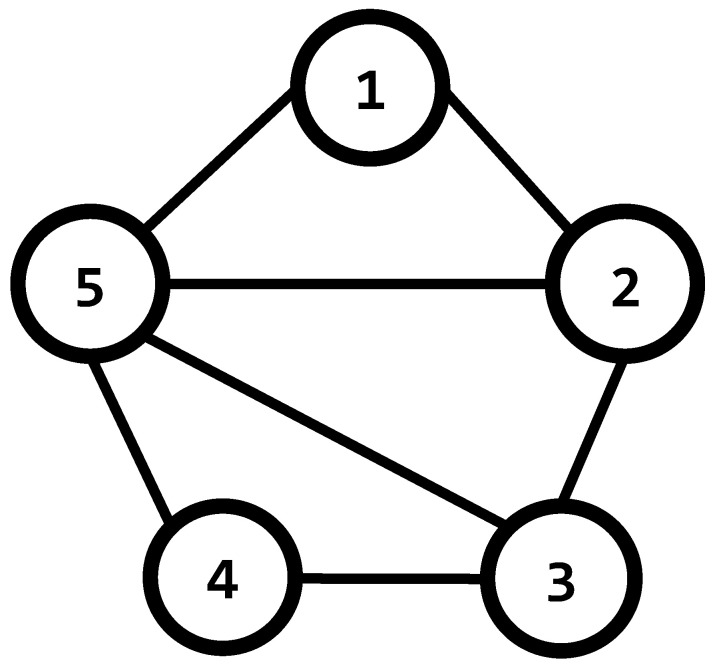
An undirected graph.

**Figure 3 entropy-25-00853-f003:**
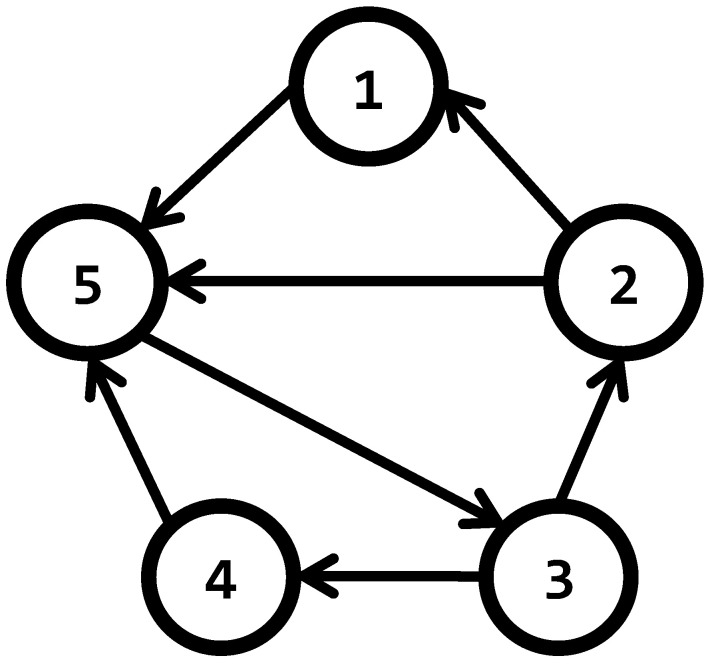
A directed graph, or digraph.

**Figure 4 entropy-25-00853-f004:**
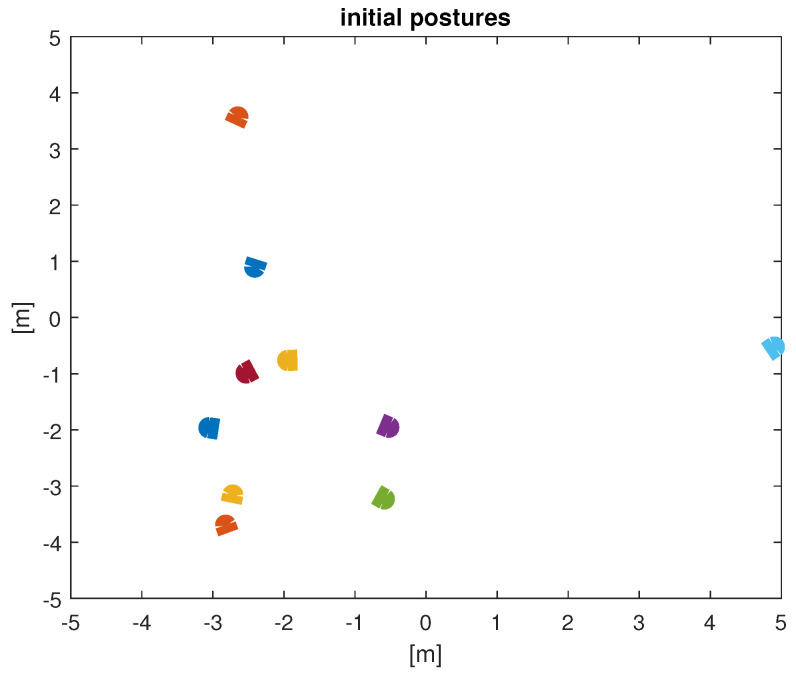
Initial positions and orientations of the 10 UAVs. Each drone is depicted as a different color.

**Figure 5 entropy-25-00853-f005:**
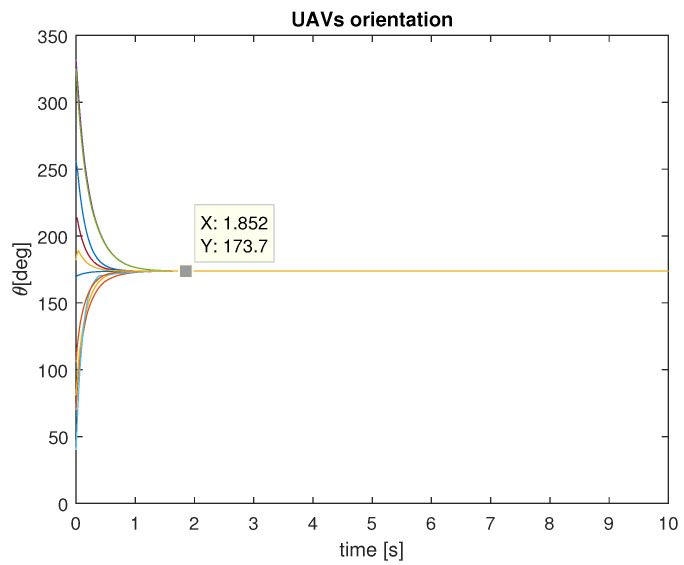
Orientations of the 10 UAVs.

**Figure 6 entropy-25-00853-f006:**
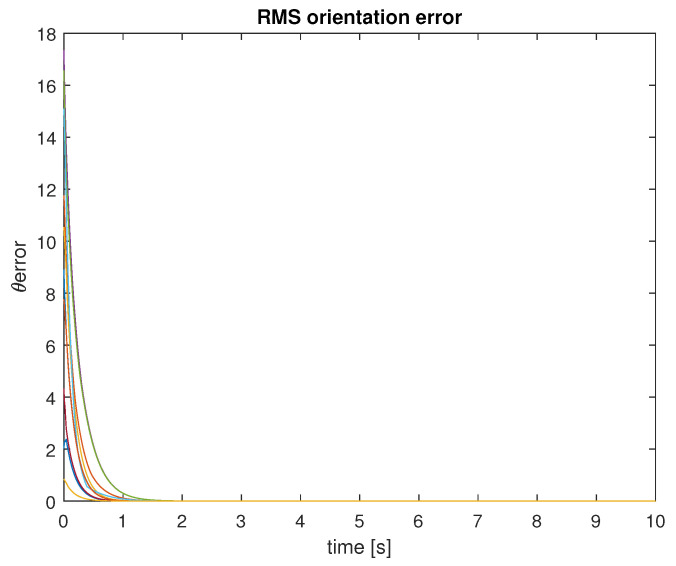
RMS orientation errors.

**Figure 7 entropy-25-00853-f007:**
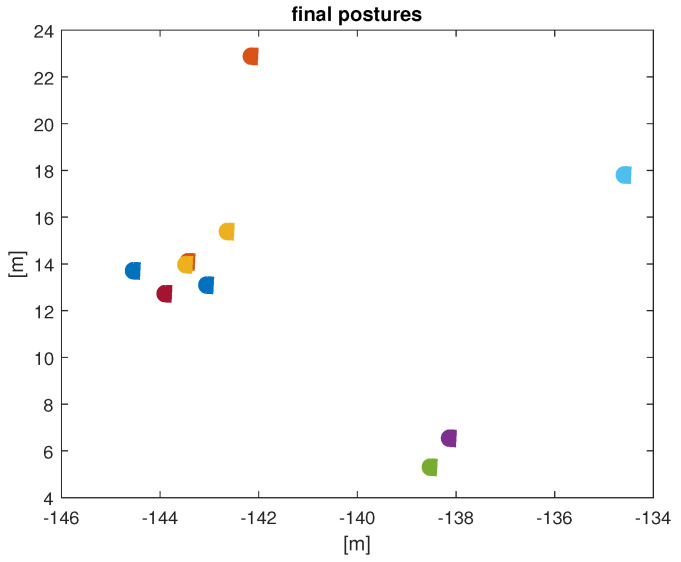
Final positions and orientations of the 10 UAVs. Each drone is depicted as a different color.

**Figure 8 entropy-25-00853-f008:**
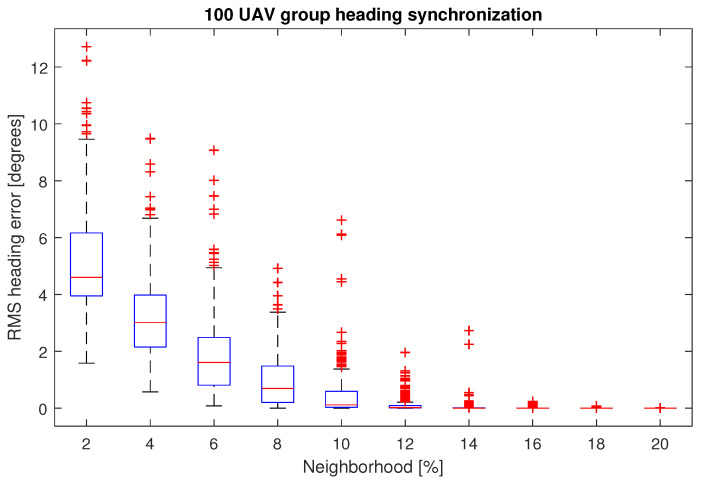
RMS heading errors for the experiment varying the size of the neighborhood over a 100-UAV swarm.

**Figure 9 entropy-25-00853-f009:**
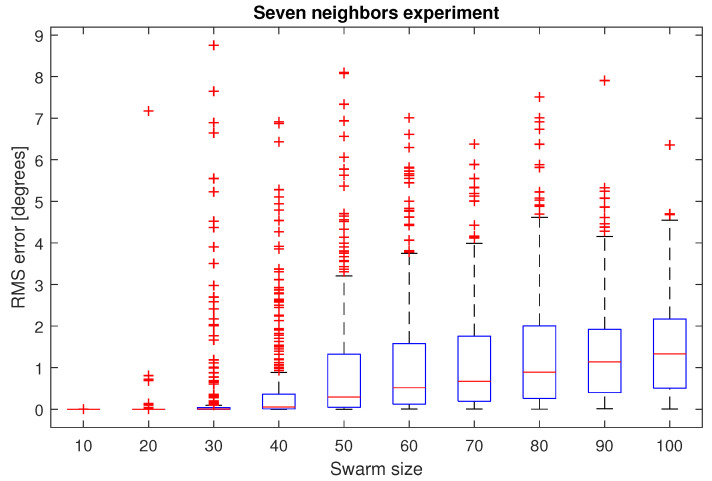
RMS heading errors for the experiment with a fixed seven-UAV neighborhood.

## Data Availability

No new data were created or analyzed in this study. Data sharing is not applicable to this article.

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
