# Peer review of "Optimum k-Nearest Neighbors for Heading Synchronization on a Swarm of UAVs under a Time-Evolving Communication Network"

_entropy, 2023, doi:10.3390/e25060853_

Round 1

Reviewer 1 Report

1.       The abstract should briefly mention the methodology used to determine the optimal neighborhood size for heading synchronization on a swarm of UAVs.

2.       In the introduction, it would be helpful to provide more context on why heading synchronization is important in flocking behaviors and how it can benefit UAV swarms.

3.       The authors should explain why they chose a P-like control algorithm for heading synchronization and how it compares to other algorithms in terms of simplicity and effectiveness.

4.       The section on graph modeling could benefit from more explanation and clarity, particularly in defining the adjacency matrix Γ and degree matrix D.

5.       It would be helpful to provide more information on how simulations were conducted, such as the specific software used and any assumptions made about the environment or UAV dynamics.

6.       The authors should clarify whether their simulation results are based on ideal conditions or if they account for potential communication failures or other disruptions that may occur in real-world scenarios.

7.       In discussing their results, the authors should provide more details on how they measured successful heading synchronization and what criteria they used to evaluate performance.

8.       The conclusion could benefit from a summary of key findings and implications for future research, rather than simply restating what was already discussed in the paper.

9.       It would be helpful to include a discussion on potential limitations or challenges associated with implementing k-nearest neighbor algorithms for heading synchronization in real-world UAV swarms.

10. literature should be enhanced in review of previous works on UAV and more wide check, as now paper seems very unaware of past works on UAV control, such examples are:
-Cao, B., Li, M., Liu, X., Zhao, J., Cao, W.,... Lv, Z. (2021). Many-Objective Deployment Optimization for a Drone-Assisted Camera Network. IEEE transactions on network science and engineering, 8(4), 2756-2764. doi: 10.1109/TNSE.2021.3057915
-Li, B., Li, Q., Zeng, Y., Rong, Y., & Zhang, R. (2021). 3D trajectory optimization for energy-efficient UAV communication: A control design perspective. IEEE Transactions on Wireless Communications, 21(6), 4579-4593. doi: 10.1109/TWC.2021.3131384
-Zhou, G., Bao, X., Ye, S., Wang, H., & Yan, H. (2021). Selection of Optimal Building Facade Texture Images From UAV-Based Multiple Oblique Image Flows. IEEE transactions on geoscience and remote sensing, 59(2), 1534-1552. doi: 10.1109/TGRS.2020.3023135
-Liu, L., Zhang, S., Zhang, L., Pan, G., & Yu, J. (2022). Multi-UUV Maneuvering Counter-Game for Dynamic Target Scenario Based on Fractional-Order Recurrent Neural Network. IEEE Transactions on Cybernetics, 1-14. doi: 10.1109/TCYB.2022.3225106
-Yang, Z., Yu, X., Dedman, S., Rosso, M., Zhu, J., Yang, J.,... Wang, J. (2022). UAV remote sensing applications in marine monitoring: Knowledge visualization and review. Science of The Total Environment, 838, 155939. doi: https://doi.org/10.1016/j.scitotenv.2022.155939

11.   Overall, the paper could benefit from clearer organization and structure, particularly in terms of separating out technical details from broader discussions of methodology and results.

Author Response

Dear reviewer, thank you for your aid in the improvement of the manuscript, attached you will find a document with the response with every recommendation given.

Reviewer 2 Report

This work focuses on a statistical analysis for determining an optimal neighborhood size for a UAVs swarm. It is an interesting work but it has different flaws and issue that need to be solved:

-         - Line 11 and all manuscript: please replace the word “robot” with something different

-         - Line 58: what kinds of UAVs have you used for these tests?

-         - Please improve introduction and description of the prelimirnaries

-         - Please improve bibliographic literature

Thank you.

Author Response

(The authors gave the same response as above.)

Round 2

Reviewer 2 Report

Thank you for improving the manuscript, now it loooks better structured concerning data and the overall presentation.